# Nursing Interventions to Reduce Health Risks from Climate Change Impact in Urban Areas: A Scoping Review Protocol

**Maria João Salvador Costa** [1,*] , **Pedro Melo** [2,*] , **Ulisses Azeiteiro** [3] , **Sara Carvalho** [3] and **Robert Ryan** [4]

1 Centre for Interdisciplinary Research in Health (Lisboa), Institute of Health Sciences, Universidade Católica Portuguesa, 1649-023 Lisboa, Portugal
2 Centre for Interdisciplinary Research in Health (Porto), Institute of Health Sciences, Universidade Católica Portuguesa, 4169-005 Porto, Portugal
3 CESAM—Centre for Environmental and Marine Studies & Department of Biology, University of Aveiro, Campus Universitário de Santiago, 3810-193 Aveiro, Portugal; ulisses@ua.pt (U.A.); saradcarvalho@ua.pt (S.C.)
4 Department of Landscape Architecture and Regional Planning, University of Massachusetts, 109 Hills North, Amherst, MA 01003, USA; rlryan@larp.umass.edu
* Correspondence: mjvcosta@ucp.pt (M.J.S.C.); pmelo@ucp.pt (P.M.)

**Abstract:** Considering that the public health sector has been considered as a key stakeholder in climate action, it seems important to understand what interventions are carried out globally by trusted professionals such as nurses engaged in health promotion and environmental health in optimizing the health of individuals, families, and communities toward the dissemination of lifestyle decarbonization and guidance on healthier climate-related choices. The objective of this review was to understand the extent and type of evidence related to the community-based interventions of nurses that are being led or have been implemented thus far with the aim of reducing the health risks from climate change impact in urban areas. The present protocol follows the JBI methodological framework. Databases to be searched include PubMed, MEDLINE complete, CINAHL, Scopus, Embase, Web of Science, SciELO (Scientific Electronic Library Online), and BASE (Bielefeld Academic Search Engine). Hand searched references were also considered for inclusion. This review will include quantitative, qualitative, and mixed methods studies from 2008 onwards. Systematic reviews, text, opinion papers, and the gray literature in English and Portuguese were also considered. Mapping the nurse led interventions or those that have been implemented thus far in urban areas may lead to further reviews that may help identify the best practices and gaps within the field. The results are presented in tabular format alongside a narrative summary.

**Keywords:** community health nursing; empowerment; patient participation; health promotion; environmental health; climate change; preventive health services; primary prevention

## 1. Introduction

Climate change affects humans globally, and its consequences contribute to an increase in diseases and ill health conditions. Therefore, climate change not only seems to be related to ecology and the environmental sciences, but is to be scientifically acknowledged and reframed as a public health issue as a result of its severe impact on human health caused by the transformation of our natural environment [1].

Recently, the Lancet and University City London (UCL) launched a joint report precisely stating that climate change is not a conspiracy theory but is instead the largest global threat of the 21st century by reviewing the likely health consequences on human societies and by revealing the directions that policymakers, practitioners, and civil society should follow in order to reverse these consequences [2].

History confirms that since 1972 in Stockholm, Sweden at the United Nations (UN) First Earth Summit, where the issue of climate change was raised for the very first time, to the Kyoto

Protocol [3], several climate declarations have been signed by several countries. Recently, with the Paris Agreement [4], the latest goal of limiting global warming by 1.5 degrees Celsius represents a new window of opportunity for all governmental institutions as well as the private sector to commit themselves to adaptation, mitigation, and resilience strategies toward carbon reduction by 2030 and net zero by 2050 [4].

Considering nursing is one of the most trusted professions in the world, nurses can be actively involved in climate action [1]. As a relevant part of the health sector, nurses may help communities in adapting to the effects of climate change as well as the risks associated with it. They play a key role in linking patients and families to their hospitals and health systems in attempts to not only reduce greenhouse gas emissions but also to influence the adoption of better lifestyles, strategies for climate efficiency in health care facilities as well as for climate preparedness in the communities [1]. We can all refer to the important role of nurses as frontline workers fighting climate change, directly or indirectly. When the world faces catastrophic disasters such as pandemics, floods, and hurricanes, amongst others, their role may be of relevance directly in the field of the disaster or, for instance, in providing care to an increased number of emergency department visits as a result of climate-related incidents and events [1].

Even when it comes to its own activities, the health care sector has to deal with their own carbon footprint, meaning that health care professionals, managers, and leaders in the world of nursing are already involved in tackling increased energy costs, stressed health services [1] increased plastic waste, and PPE (personal protective equipment) as one of the environmental side effects of COVID-19 [5].

As an example, the world is currently facing a new trend regarding the production and use of plastics, and some studies have already referred to an increase in plastic debris by 2030, causing health and environmental concerns to the health care sector, which is still dealing with the challenging situation of trying to reduce plastic leakage and pollution [5]. The authors have acknowledged within the literature a call to action by nurses to address climate change, as communities feel that this as an emergent health concern. It appears that as a collective discipline, nursing is now starting its way forward in tackling climate change as a global crisis [6]. Despite the fact that nurses around the world may feel that they are still poorly prepared to get involved with climate action, it seems to represent a moral imperative for planetary health, within the scope of their profession [6]. However, to address this issue, nurses must be confident that they can engage in coordinated and collaborative paths within their nursing practice [6]. Although nurses have already been living through a complex and overburdened health care system, especially with the recent COVID-19 pandemic, the extent of the health risks and effects of our changing climate seem undeniable, and therefore there is a professional obligation to engage in climate change mitigation and adaptation [6].

If health institutions and professionals, namely, nurses, are identified as trusted figures in whom the public tends to rely on, they may act as key players in educating the public in general, but also their own peers, on climate literacy and strategies to help the health sector reduce its emissions and controlling the climate health impact [1].

Community and public health specialist nurses (CPHSNs) are an example of qualified professionals to lead as influencers on climate action, empowering communities as required in actively engaging all individuals in achieving socio-political change [7], meaning that they will intervene to improve the community, offering prospects for the citizens to participate and own their own lives and health [8].

Health practitioners such as CPHSNs are fundamental in health promotion, which drives the public health sector, and have recently been considered as key stakeholders when it comes to tackle climate change. CPHSN regularly engage with and lead individuals, groups, families, communities, and governmental institutions, supporting healthier lifestyle choices, advocating for adequate decisions from policymakers in urban settings, and reducing the global carbon footprint [9].

CPHSNs identify and diagnose environmental hazards as a result of climate change impact in urban areas and determine adequate interventions that might lead to behavioral changes either by inspiring, educating, or raising awareness of the consumers' conduct. Nurses worldwide are currently following the United Nations recommendations for climate action within cities and influencing communities: nurses help individuals and families to either mitigate their negative environmental impact such as decarbonizing lifestyles or to adapt to the climate change impact by increasing their resilience and, for instance, reclaiming green spaces in the cities. Either way, and most importantly, nurses will be reducing the health risks associated with climate change impact [10].

Carvalho, Bisquert i Perez, Cartea, and Azeiteiro [11] presented diet and climate change associated with environmental education and lifestyle decarbonization during the IV Resclima International Seminar at Santiago de Compostela in Spain. These authors argue that, somehow, history explains the different agrifood systems developed for the communities 'supply, having adapted to the particularities of each territory, resulting in a variety of diet models worldwide. Nonetheless, since the Industrial Revolution, fossil fuels and pesticides have globally transformed what we know as traditional agriculture, changing the methods of production, distribution, and commercialization around the world. Consequently, this has changed the dietary patterns in many regions and amongst populations to unsustainable practices, leading to a massive impact not only on the planet's environment, with the emission of higher levels of greenhouse gases (GHG) into the atmosphere, but also on people's health. According to Carvalho, Bisquert i Perez, Cartea, and Azeiteiro [11], and referring to data from the Intergovernmental Panel on Climate Change (IPCC), twenty five percent of GHG originate from agriculture, forestry, and soil use, so it urges the promotion of local and regional alternatives and the education of populations into decarbonized and sustainable diets such as a reduction in dairy consumption and animal proteins. This totally aligns with the Final Report of the High-Level Panel of the European Decarbonization Pathways Initiative, a package of measures to decarbonize lifestyles, developed by the European Commission, which confirms that the transition from a beef-based meal to a plant-based one cuts emissions by around 10 times [12].

Further aligned with the Paris Agreement [4], the package of measures of the European Commission, as above-mentioned, clearly links the need to reduce the global carbon footprint to neutral levels by 2050 to the concept of "livable cities", where behavior change strategies and social innovations are required from policymakers, politicians, and private businesses for replication and dissemination [12].

Planning for social innovation may involve reclaiming green infrastructures and sustainable designs for climate resilient cities [13] such as greenways, originally defined in the Lille Declaration, in 2000, which offer communities healthier lives by improving non-motorized routes, encouraging closer relationships between citizens and simultaneously mitigating the pollution in the cities [14]. Greenways, as vegetated natural areas that connect people and places, seem to be growing around the world for its natural, recreational, and cultural purposes [15]. Hence, developing infrastructures such as these as well as encouraging their use as health resources will certainly reduce the impact of climate change and increase the health status of communities.

Therefore, whilst planning, designing, constructing, or reshaping health care facilities, sustainability principles must be acknowledged, which means incorporating landscaped environments that contribute to reduced energy use and waste as well as GHG emissions. For instance, planting trees or native vegetation and using sunshades can contribute to reducing the *urban island heating effect* of a hospital. Additionally, pedestrian and bicycle access between buildings can and should be planned, avoiding car use inside or near the grounds of the facilities [1].

Nevertheless, regardless of how committed these health care professionals might be with mitigating the climate crisis, our planet seems sentenced to warm up between 1.4 and 4.3 °C by the end of the 21st century, hence why bottom–up approaches that prepare the public and their communities to best adapt to climate change are also extremely significant.

Adaptation strategies are therefore vital in preparing resilient communities, a reason why nurses may anticipate the health needs of communities [1] by caring and addressing each one as a nursing care unit, as considered by the Model of Assessment, Intervention, and Community Empowerment (MAIEC) [16].

According to this theoretical model (MAIEC), if communities are to be considered as *nursing care units*, nurses become key in enhancing the health status of communities by assessing, identifying, and diagnosing health risks due to compromised processes, compromised participation, or compromised leadership in relation to climate change impact. As a result, nurses at the forefront of the public health sector will be very well-placed to implement the necessary interventions and help reduce unwanted health risks from climate change impact in urban areas as a process and as a result. However, and preferably establishing partnerships with other stakeholders such as local health and government officials, nurses can enhance their work and influence, monitoring the most vulnerable by assessing the health threats (such as infectious diseases), health behaviors, mortality, or morbidity rates within communities, and drafting their responses and interventions (such as email or phone alerts on how to best deal with extreme heat and specific air pollution events) or plan their emergency preparedness [1].

Environmentally concerned nurses are getting together worldwide to meet the health and climate challenges. Examples of such groups of nurses are Health Care without Harm (HCWH) and other nursing associations such as the American Nurses Association (ANA) and the International Council of Nurses (ICN), highlighting the importance of nursing in assisting with mitigation and adaptation strategies, and developing climate related national policies and action plans.

Therefore, to support current and future research, it seems of relevance to gather existing evidence on nursing interventions toward climate change mitigation, adaptation, understanding what is currently being advocated by policy making.

For this purpose, a preliminary search on PubMed and MEDLINE Complete was conducted, and no current or in-progress systematic reviews or scoping reviews were found that map the existing evidence on nursing interventions to reduce the health risks from climate change impact, specifically in urban areas.

Although Sayre, Rhazi, Carpenter, and Hughes [1] have partially approached the topic, they do not provide any mapping of evidence on such nursing interventions in urban areas, the reason why we will conduct this scoping review with the objective to map published and unpublished research on this emergent field of intervention. The present protocol will be registered on the OSF registries platform, and an abstract will follow for publication.

## 2. Review Question

The present review seeks to respond to the following question:

What community-based interventions are being led or implemented by nurses to reduce health risks from climate change impact in urban areas?

## 3. Eligibility Criteria

The type of studies to be included in the review will acknowledge the following PCC framework.

### 3.1. Participants

This review will consider all studies related to *nurses* involved in promoting community-based interventions using climate-related mitigation or adaptation strategies with the aim at reducing health risks from climate change impact in urban areas.

### 3.2. Concept

This review will consider all papers and documents discussing or exploring the nurses' community-based interventions using climate-related mitigation or adaptation strategies, with the aim to reduce the health risks from climate change impact in urban areas.

### 3.3. Context

This review will only consider studies including urban contexts and urban communities (including hospital settings as communities and therefore considered as nursing care units) [17], that meaning rural contexts will be excluded from this review.

### 3.4. Types of Sources

Quantitative, qualitative, mixed methods studies will be considered as well as text and opinion papers and other gray literature such as worldwide theses and dissertations published in repositories. Other relevant sources such as documents or reports obtained through the authors' professional networks and social media will also be considered.

## 4. Methods

### 4.1. Search Strategy

This review will follow the JBI methodology for scoping reviews, in accordance with the Preferred Reporting Items for Systematic Reviews and Meta-Analyses extension for Scoping Reviews (PRISMA-ScR) checklist [18], and aims to locate both published and unpublished studies. The search strategy included several steps, as described below:

1st Step—Applying a three-step methodology, the authors have initially conducted a preliminary search on PubMed and MEDLINE Complete by identifying relevant articles on the topic (Appendix A).

2nd Step—In order to develop a full search strategy, search terms (Appendix B), keywords, and MeSH descriptors (Appendix C) contained in the titles and abstracts of relevant articles on the topic were considered.

3rd Step—Screening of all reference lists from the articles to add further relevant studies that would have been missed otherwise (snowballing) will also be considered.

Although several nursing and health organizations have already called out for nursing climate action such as the Australian College of Nursing (CAN), the Canadian Nurses Association (CAN), the Royal College of Nursing (RCN), and the Alliance of Nurses for Healthy Environments (ANHE) [6], the authors will only focus on the most contemporaneous articles published from 2008, since the American Nurses Association (ANA) adopted a resolution on global climate change, offering acknowledgement of the challenges the phenomenon poses to the world and providing additional guidance for nurses and the need for advocacy for change at the individual and policy levels as well as the need to promote nursing knowledge on the relationship climate change/population health.

A full search will be conducted on the following databases: PubMed, Medline Complete, CINAHL, Scopus, Embase, Web of Science, SciELO (Scientific Electronic Library Online), and BASE (Bielefeld Academic Search Engine). Gray literature from hand searched references and unpublished studies will also be considered for inclusion.

A word of clarification is required to state that regarding PubMed and Medline, when tested simultaneously, although there is an expectation of retrieving duplicates, both are not a total duplication of each other as Medline Complete also accesses around 400 journals that, while not integrated in PubMed/Medline, have been vastly validated by the biomedical community.

### 4.2. Study/Source of Evidence Selection

All relevant studies identified from the search and from 2008 onward in Portuguese and English will be assessed in detail by two or more independent reviewers against the inclusion criteria and the relevant papers will be retrieved in full. Citation details will be imported into the JBI System for the Unified Management, Assessment, and Review of Information (JBI SUMARI; JBI, Adelaide, Australia) [19]. The search process will be fully reported in the final scoping review and presented in a Preferred Reporting Items for systematic Reviews and Meta-Analysis (PRISMA) checklist [18] to clarify the course of action taken throughout. The records retrieved will be uploaded into Rayyan [20]—a free

online version to manage all selected results. By using Rayyan, a blind selection process will be carried out by two or more chosen researchers and a supplementary element of the team will be called out, allowing for any disagreements to be resolved through discussion. Rayyan is a web-tool used for a comprehensive analysis of the studies selected, assessed, and retrieved, easily deleting duplicated records, and also enabling the resolution of each one of the others, according to the inclusion criteria.

The results of the search and the study inclusion process will result in a full report in the scoping review presented in a Preferred Reporting Items for Systematic Reviews and Meta-analysis extension for a scoping review flow diagram (PRISMA-ScR) [18].

All records retrieved can then be exported to a free online version of Zotero [21], a web references management tool.

In the review, the authors will summarize the reason as to why some papers, not meeting the inclusion criteria have, therefore, been excluded.

*4.3. Data Extraction*

The authors have currently developed a draft data extraction tool designed to retrieve evidence extracted from papers included in the scoping review by two or more independent reviewers. The data extracted will include specific details about the participants of each study, the nursing community-based interventions identified and aimed at reducing health risks from climate change, the type of context in which they took place, the study methods used, and other key findings that the researchers might find relevant to answer the review question.

Provided below, the draft data extraction tool supports the objective and the aim of the present review question, and shows the key characteristics of the selected studies (see Appendix D). However, it can and surely will be enhanced throughout the review process, ensuring that all changes performed will be fully described in the scoping review.

To complete missing data, the authors of papers may be contacted if required.

## 5. Data Analysis and Presentation

Although it is not possible to predict the exact outcomes of a proposed review, the authors have high expectations on the key contributions that nurses may bring toward the mitigation and adaptation to climate change. The authors of the present manuscript also expect that this review may encourage further studies in the field and lead to the implementation of new public health programs and policies that recognize the importance of the complex interventions of the nursing workforce.

The authors will present the review results by using a tabular form (data analysis tables), and a narrative summary will accompany the tabulated results, as required, to provide a detailed description to the reader, clarifying the main findings and relating the results to the review objective and question.

The authors have considered that this scoping review protocol represents an opportunity to plan how to best present their data search, extraction and analysis process, and add the necessary transparency to the whole research, required by the reader [19]. Enabling the identification of the authors of each paper, year of publication, country (where the study was conducted, study details (e.g., aims, study design, study population, sample size), study setting, and nursing community-based interventions targeted to climate change mitigation and adaptation (see Appendix D) will hopefully answer the review question, which is extremely relevant when starting to plan new global public health policies using a nursing-centered approach.

**Author Contributions:** Conceptualization, M.J.S.C.; Methodology, M.J.S.C.; Validation, P.M., U.A. and R.R.; Investigation, M.J.S.C., P.M., U.A., S.C. and R.R.; Resources, M.J.S.C., P.M., U.A., S.C. and R.R.; Visualization, M.J.S.C.; Project administration, M.J.S.C.; Funding Acquisition N/A.; Writing—original draft preparation, M.J.S.C.; Writing—review and editing, M.J.S.C., P.M., U.A. and R.R. All authors have read and agreed to the published version of the manuscript.

**Funding:** This research received no external funding.

**Institutional Review Board Statement:** Not applicable.

**Informed Consent Statement:** Not applicable.

**Data Availability Statement:** For data supporting reported results please contact the authors of this review.

**Acknowledgments:** The authors wish to thank Maria Perdigão from the Library Services at Universidade Católica Portuguesa, Lisbon Campus for all of the support with the use of databases in the present manuscript.

**Conflicts of Interest:** The authors declare no known conflicts of interest.

## Appendix A. Preliminary Search

| Database | Keywords and Search Words | Records Retrieved (N) |
|---|---|---|
| PubMed | ((climate change[MeSH Terms]) AND (((((((empowerment[MeSH Terms]) OR ("patient participation"[MeSH Terms])) OR ("health promotion"[MeSH Terms])) OR ("environmental health"[MeSH Terms])) OR ("public health"[MeSH Terms])) OR ("Preventive Health Services"[MeSH Terms])) OR ("Primary Prevention"[MeSH Terms]))) AND ((nursing[MeSH Terms]) OR ("community health nursing"[MeSH Terms])) | 16 |
| Medline complete | (MH "climate change") AND (MH ((empowerment OR "patient participation" OR "health promotion" OR "environmental health" OR "public health" OR "Preventive Health Services" OR "Primary Prevention"))) AND (MH ((nursing OR "community health nursing"))) | 1 |

## Appendix B. Search Terms and Scope Notes

| Search Terms | Scope Notes |
|---|---|
| Climate Change | Any significant change in measures of climate (such as temperature, precipitation, or wind) lasting for an extended period (decades or longer). It may result from natural factors such as changes in the Sun's intensity, natural processes within the climate system such as changes in ocean circulation, or human activities. |
| Empowerment | Process of increasing the capacity of individuals or groups to make choices and to transform those choices into desired actions as deigned by the individuals or groups. |
| Patient participation | Patient involvement in the decision-making process in matters pertaining to health. |

| | |
|---|---|
| Community health nursing<br><br><br>Health promotion | General and comprehensive nursing practice is directed at individuals, families, or groups as it relates to and contributes to the health of a population or community. This is not an official program of a Public Health Department. Encouraging consumer behaviors most likely to optimize health potentials (physical and psychosocial) through health information, preventive programs, and access to medical care. |
| Environmental health | The science of controlling or modifying those conditions, influences, or forces surrounding man which relate to promoting, establishing, and maintaining health. |
| Preventive Health Services | Services designed for HEALTH PROMOTION. |
| Primary Prevention | Specific practices for the prevention of disease or mental disorders in susceptible individuals or populations. These include HEALTH PROMOTION including mental health; protective procedures such as COMMUNICABLE DISEASE CONTROL; and monitoring and regulation of ENVIRONMENTAL POLLUTANTS. Primary prevention is to be distinguished from SECONDARY PREVENTION and TERTIARY PREVENTION. |

## Appendix C. Keywords and MeSH Descriptors

| Keywords | Descriptors MeSH |
|---|---|
| Community Empowerment | Community Health Care, Ecological Community, Empowerment, Community Care network, Community services, Community Health Education, Community Health Systems, Patient Empowerment, Community Development |
| Climate Change | Climate Change<br>Climate Process<br>Climate control |
| Community Health Nursing | Health Visitors, Home Nurses, Community Health Care, Health Services, Community Health Education, Community Health Systems, Community Health Networks, Home Health Care, Health Centers |
| Public Health Nursing | Public Health Schools<br>Public Health students<br>Community Health<br>Preventive Medicine and Public Health<br>Environment<br>Public Health Service<br>Home Nurses<br>Public Health Education for Professionals<br>Community Health<br>Public Health Practice |

**Appendix D. Data Extraction Instrument**

| Scoping Review Details | |
|---|---|
| Scoping Review title | Nursing interventions to reduce health risks from climate change impact in urban areas—A scoping review |
| Review Objective | To map nursing interventions that reduce health risks from climate change impact in urban settings. |
| Review Question | Which nursing interventions have been led/implemented so far to reduce health risks from climate change impact in urban areas? |

| **Inclusion/Exclusion criteria** |
|---|
| Population |
| Concept |
| Context |
| Types of evidence source |
| **Evidence Source details and characteristics** |
| Citation details (author, date, title, journal, volume, issue, pages) |
| Country of origin |
| Method |
| Aim of the study |
| Participants (age, sex, number) |
| Study setting |
| **Results extracted from source of evidence (in relation to the concept of the review)** |
| Number of items in tool |
| Nursing interventions identified |

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
