# Peer review of "Nursing Interventions to Reduce Health Risks from Climate Change Impact in Urban Areas: A Scoping Review Protocol"

_nursrep, doi:10.3390/nursrep13010045_

Round 1

Author Response

Reviewer 1

Nursing interventions to reduce health risks from climate change impact in urban areas: a Scoping Review Protocol This is a preview of a planned systematic review aimed at: will be mapping nurses interventions led or implemented so far with the aim of reducing health risks from climate change impact in urban areas. The authors of the protocol underlined that the Public Health sector has been considered a key stakeholder in climate action. As a relevant part of the health sector, nurses may help communities in adapting to the effects of climate change as well as the risks associated with it. Therefore, it was considered necessary to understand what interventions are carried out worldwide by the nurses, engaged in health promotion and environmental health, and optimizing health.

Detailed comments:

Point 1 - The introduction to the study is interesting but too extensive and some passages deviate from the topic. I suggest shortening the paragraphs from line 117 to line 137.

Response to Point 1 – Thank you for your comment and suggestion. We too agree with the relevance of the subject area and feel very enthusiastic with the study. The more we learn, the more we wish to share with the wider academic community. We have seen you have considered that we provided sufficient background and included all relevant references. However, MDPI has required a minimum of 4000 words for the main text, even considering this is just a scoping protocol, therefore we assumed we needed to be even more thorough when introducing the subject to the reader and relating it with the need for understanding what is currently being done to mitigate and adapt to climate change, namely by nurses, who develop complex interventions towards their patients, families and communities. Nevertheless, we will take into account your suggestion and synthetize the paragraphs above mentioned.

Point 2 - The review study protocol presents transparently the data collected by the authors as well as the planned research process. The results of the research may be important in the planning of pre- and postgraduate education of nurses who will be addressing public health problems as a result of climate change.

Response to Point 2 – Dear reviewer, we thank you for your constructive comments. We appreciate that you could assess the transparency of the research process as we wish to be fully compliant with the research ethics requirements. Furthermore, we thank you for your kind words by acknowledging the potential interest of the study for the future of curriculum design in nursing. We feel humbled and thankful by your words. We all wish to contribute for the enhancement of nursing education and practice in regards to Public Health, key in our global community.

Once again, thank you for your input and encouragement.

We will ensure the improvement of our manuscript by taking in consideration all of your recommendations.

Kind Regards

The authors

Reviewer 2 Report

The authors (AA) aim to summurize the current evidend about community-based interventions led or implemented by nurses to reduce health risks from climate change impact in urban areas. This could be a review useful to increase our knowledge of the issue. Addressing all the issues below reported could make this manuscript eligible for the publication.

Introduction:

The introduction is scattered and does not make the key points clear to the reader. AA should be synthesised this section and add the adequate references in some points such as lines 51-57.

Methods:

What are the main outcomes that AA will consider in this scoping review?

In the paragraph 3, AA should better explain the expected results and data presentation.

Appendix II:

Why is climate change not present among the search terms?

Author Response

Reviewer 2

1.The authors (AA) aim to summarize the current evidence about community-based interventions led or implemented by nurses to reduce health risks from climate change impact in urban areas. This could be a review useful to increase our knowledge of the issue. Addressing all the issues below reported could make this manuscript eligible for the publication.

Response to Point 1 – Dear Reviewer, the authors of the present manuscript appreciate your comments and recommendations. We too hope this study will be a contribute to increase our knowledge of the issue. We will ensure that we address all the issues reported by you in order to be eligible for publication.

2.Introduction:

The introduction is scattered and does not make the key points clear to the reader. AA should be synthesised this section and add the adequate references in some points such as lines 51-57.

Response to Point 2 – Dear Reviewer, we feel sorry that you didn´t find the introduction clear enough. We have attempted to show to the reader how climate change affects humans globally and how the phenomenon is an emergent public health issue.

The fact that recent studies have found that the Public Health sector seems to play a key role in climate change mitigation and adaptation, we are attempting to map what nursing interventions are currently being led or implemented around the world in that direction, considering nurses are central players of the Public Health sector, one of the most important stakeholders in tackling this global environmental crisis.

MDPI requires 4000 words for the main text, reason why we assumed the introduction should be more thorough and comprehensive. However, we will review it again, according to your recommendations and see how we can make both introduction and its key points more explicit.

We will also ensure paragraphs such as the above-mentioned have the right references associated. In lines 51-57 the reference is [1], accidentally omitted.

3.Methods:

What are the main outcomes that AA will consider in this scoping review?

In the paragraph 3, AA should better explain the expected results and data presentation.

Response to Point 3 – Dear Reviewer, thank you for your query. In fact, although this is a scoping protocol, including a plan for a review, writing more about the expected results may actually help establishing the relevance of our study.

As it is not possible to predict the exact outcomes of the proposed review, we assumed this would only lead to a lack in accuracy. Nevertheless, we acknowledged your recommendation, and we will ensure to extend sub-chapter 3 and specifically clarify our high expectations on the key contributes that nurses may bring towards the mitigation and adaptation to climate change. We also expect that the outcomes of the present study may encourage further reviews and lead to the future implementation of new public health programs and policies that recognise the importance of complex interventions of the nursing workforce.

4.Appendix II: 

Why is climate change not present among the search terms?

Response to Point 4 – Dear Reviewer, you are correct. Climate change is one of the relevant keywords and search terms for the present study, however inadvertently it has been misplaced, not appearing in Appendix II. That has now been amended. Thank you.

We appreciate your time and recommendations, hoping that addressing all issues found, the manuscript may be finally eligible for publication. We will shortly re-submit the revised manuscript accordingly, as required.

Again, thank you.

Kind Regards

The authors

Reviewer 3 Report

First of all, I would like to congratulate the authors for their interest in such a topical, pertinent, and extremely important topic.

The manuscript presented is a scoping review protocol, following the method proposed by the JBI, with the objective to map community-based interventions, led or implemented by nurses, to reduce health risks from climate change impact in urban areas.

A previous search was carried out to verify if there was already a systematic review or scoping review published or in progress. Because it does not exist, the authors registered the protocol at the OSF.

The introduction frames the issue of climate change and the role of nurses in this area.

The "PCC" review question is presented in the introduction. It is suggested that review question be a topic separate from the introduction (for example: 2. Review Question)

In sub-section 1.2 (it is suggested to change "1.2. Inclusion Criteria" to "3. Eligibility Criteria"): The eligibility criteria are presented in relation to participants, concept, and context; The types of studies to be included in the review are suitable for a scoping review. The authors present here the temporal limiter of publication (and the justification), but it is suggested that they change this to the "search strategy" section (as they did with regard to the absence of the "language of publication" limiter).

In the method: the steps that precede a scoping review were completed, to identify the relevant search terms; two search strategies are presented (in appendix), the selection process is explained, the extraction process is explained (and the data extraction tool is presented).

The "expected results and data presentation" section explains how the results will be presented.

It would be good to clarify:

> the abstract reads "Databases to be searched include PubMed, MEDLINE complete, CINAHL, Scopus, Embase, Web of Science, SciELO (Scientific Electronic Library Online and Repositório Científico de Acesso Aberto de Portugal (RCAAP)", but in the "information sources" section they don't mention RCAAP.

> why limit the search to RCAPP? RCAAP is a Portuguese open access initiative and aims to store, preserve and promote access to scientific knowledge produced in Portugal. By limiting the search to a Portuguese repository, the authors may be excluding the possibility of including theses and dissertations from other possibly relevant countries. Why not use a database that contains theses and dissertations from a larger geographical area, such as BASE (https://www.base-search.net/)?

> the selection of studies will be carried out using Rayan (which allows a blind selection process). How many researchers will carry out the articles selection process and how will the authors proceed in case of a tie?

> Given that MEDLINE is a database that is integrated into PubMED, does it make sense to present both search strategies (MEDLINE strategy and PubMed Strategy)? Certainly, the results presented by the MEDLINE search are also presented in the PubMED search.

> in the section "expected results and data presentation" the expected results with this scoping are not presented. Does it make sense to have "expected results" in the section name? Do the authors expect some results? 

Bearing in mind that the manuscript refers to a scoping review protocol, and the Nursing Reports journal is flexible with regard to the sections to be presented, it is suggested that the manuscript be divided with the sections proposed by JBI (https://jbi.global /sites/default/files/2022-02/JBI_Protocol_Template_Scoping_Reviews.docx).

Author Response

Dear Reviewer,

We would like to thank you for your constructive feedback and comments. The authors will respond below, point by point, based on the comments made.

First of all, I would like to congratulate the authors for their interest in such a topical, pertinent, and extremely important topic. Thank you for your acknowledgement.

The manuscript presented is a scoping review protocol, following the method proposed by the JBI, with the objective to map community-based interventions, led or implemented by nurses, to reduce health risks from climate change impact in urban areas.

A previous search was carried out to verify if there was already a systematic review or scoping review published or in progress. Because it does not exist, the authors registered the protocol at the OSF.

The introduction frames the issue of climate change and the role of nurses in this area.

The "PCC" review question is presented in the introduction. It is suggested that review question be a topic separate from the introduction (for example: 2. Review Question)

Response: Thank you for your recommendation, the authors will amend the manuscript as recommended.

In sub-section 1.2 (it is suggested to change "1.2. Inclusion Criteria" to "3. Eligibility Criteria"): The eligibility criteria are presented in relation to participants, concept, and context; The types of studies to be included in the review are suitable for a scoping review. The authors present here the temporal limiter of publication (and the justification), but it is suggested that they change this to the "search strategy" section (as they did with regard to the absence of the "language of publication" limiter).

Response: Thank you for your recommendation, the authors will take this in consideration and amend the manuscript as recommended.

In the method: the steps that precede a scoping review were completed, to identify the relevant search terms; two search strategies are presented (in appendix), the selection process is explained, the extraction process is explained (and the data extraction tool is presented).

The "expected results and data presentation" section explains how the results will be presented.

Response: Thank you.

It would be good to clarify:

> the abstract reads "Databases to be searched include PubMed, MEDLINE complete, CINAHL, Scopus, Embase, Web of Science, SciELO (Scientific Electronic Library Online and Repositório Científico de Acesso Aberto de Portugal (RCAAP)", but in the "information sources" section they don't mention RCAAP.

Response: In fact, you are correct. Accidentally, this has been omitted from the text, it will be amended as a matter of priority.

> why limit the search to RCAPP? RCAAP is a Portuguese open access initiative and aims to store, preserve and promote access to scientific knowledge produced in Portugal. By limiting the search to a Portuguese repository, the authors may be excluding the possibility of including theses and dissertations from other possibly relevant countries. Why not use a database that contains theses and dissertations from a larger geographical area, such as BASE (https://www.base-search.net/)?

Response: Thank you for your pertinent comment. We will take this in consideration and test the suggested database to assess if any relevant studies may be retrieved.

> the selection of studies will be carried out using Rayan (which allows a blind selection process). How many researchers will carry out the articles selection process and how will the authors proceed in case of a tie?

Response: Again, thank you for highlighting such important detail. In fact, for some reason, we have mentioned on lines 234-237 the use of a blind process, using Rayyan, however this has not been thoroughly explained in what concerns to the researcher’s role. Therefore, you are happy to clarify and add to this paragraph that by using Rayyan, a blind selection process will be carried out by two of the researchers and a third element of the team will be called out, allowing any disagreements to be resolved through discussion.

> Given that MEDLINE is a database that is integrated into PubMED, does it make sense to present both search strategies (MEDLINE strategy and PubMed Strategy)? Certainly, the results presented by the MEDLINE search are also presented in the PubMED search.

Response: Dear reviewer, we have acknowledged your concern, however when preparing the search equations to test the databases we have received bibliographical advice and support from our library services at Universidade Católica Portuguesa who have confirmed to us that by searching on PubMed we are including more than 5 000 scientific journals indexed on Medline, as well as all registers integrated on Pubmed Central.

Medline Complete (usually referred to as Medline) is a database that accesses around 1800 journals indexed to Medline (and PubMed), but also accesses around 400 journals that, not integrated in PubMed/Medline, have been vastly validated by the biomedical community. Summarizing, in regards to PubMed and Medline, when tested simultaneously, although there is an expectation for retrieving duplicates, both are not a total duplication of each other. We hope this makes now more sense, as this has been the reason why we have planned our search the way we did.

> in the section "expected results and data presentation" the expected results with this scoping are not presented. Does it make sense to have "expected results" in the section name? Do the authors expect some results?

Response: Thank you for your query. In fact, it appears that we have solely focused on the data presentation details, however we are currently receiving reviewer´s recommendations to clarify the expected results, despite the present manuscript being merely a protocol. We are learning that writing further about the expected results may actually help confirming the relevance of our study.

As it is not possible to predict the exact outcomes of the proposed review, we have assumed this would only lead to a lack in accuracy. Nevertheless, we are now acknowledging all recommendations, and we will ensure to extend sub-chapter 3 and specifically clarify our high expectations on the key contributes that nurses may bring towards the mitigation and adaptation to climate change. We also expect that the outcomes of the present study may encourage further reviews and encourage studies that may contribute to the implementation of new public health programs and policies that recognise the importance of complex interventions of the nursing workforce.

Bearing in mind that the manuscript refers to a scoping review protocol, and the Nursing Reports journal is flexible with regard to the sections to be presented, it is suggested that the manuscript be divided with the sections proposed by JBI (https://jbi.global /sites/default/files/2022-02/JBI_Protocol_Template_Scoping_Reviews.docx).

Response: In fact, the authors have chosen a protocol for a scoping review as it has been defined by JBI Manual for Evidence Synthesis (2020)1 to “map the key concepts that underpin a field of research, as well as to clarify working definitions, and/or the conceptual boundaries of a topic (Arksey & O’Malley 2005)2.

The protocol, as well as the method recommended by JBI Manual for Evidence Synthesis, has been strictly followed, however the identification of sections and sub-sections has indeed been adapted to Nursing Reports/MDPI publication requirements. Nevertheless, the authors of the present manuscript are willing to revert this back to the JBI original format, if recommended by you, and fully accepted by the publisher.

Other than that, any deviation from the protocol will always be clearly explained in the scoping review report and in future publications that may follow.

We truly appreciate your time and recommendations, hoping that addressing all issues found, the manuscript may be finally eligible for publication. We will shortly re-submit the revised manuscript accordingly, as recommended.

Again, thank you for your time and guidance.

1 Peters MDJ, Godfrey C, McInerney P, Munn Z, Tricco AC, Khalil, H. Chapter 11: Scoping Reviews (2020 version). In: Aromataris E, Munn Z (Editors). JBI Manual for Evidence Synthesis, JBI, 2020. Available from https://synthesismanual.jbi.global. https://doi.org/10.46658/JBIMES-20-12

2Arksey, H., & O'Malley, L. (2005). Scoping Studies: Towards a Methodological Framework. International Journal of Social Research Methodology: Theory & Practice, 8(1), 19–32. https://doi.org/10.1080/1364557032000119616

Kind Regards

The authors

Reviewer 4 Report

Dear authors.

Thank you for the opportunity to review this paper and congratulations on your work.

The manuscript aims to present the protocol of a scoping review to investigate nursing interventions to reduce health risks from the impact of climate change in urban areas.

The abstract and the theoretical framework are very well explained, with a large number of references that help to introduce and understand the topic to be investigated. The study design as well as the Keywords and MeSH descriptors are correct. The protocol is well-structured and complete. It is a very complete article, and interesting to read, although I have some questions:

a) By the MeSH descriptors Public Health Nursing and Community Health Nursing is it possible to discriminate studies that are conducted by nurses? Or is there a risk of loss of articles?

b) Will inter-operator reliability be studied?

Congratulations on your work.

Best regards.

Author Response

Reviewer 4

Dear authors.

Thank you for the opportunity to review this paper and congratulations on your work.

Response - Thank you for your kind words.

The manuscript aims to present the protocol of a scoping review to investigate nursing interventions to reduce health risks from the impact of climate change in urban areas.

The abstract and the theoretical framework are very well explained, with a large number of references that help to introduce and understand the topic to be investigated. The study design as well as the Keywords and MeSH descriptors are correct. The protocol is well-structured and complete. It is a very complete article, and interesting to read, although I have some questions:

  1. By the MeSH descriptors Public Health Nursing and Community Health Nursing is it possible to discriminate studies that are conducted by nurses? Or is there a risk of loss of articles?

Response to Point 1 – Dear reviewer, we are grateful and humbled with your thorough compliments on our study and we thank you for your query.

We believe we could narrow down the search to solely identify studies conducted by nurses, however, and quoting your words, there would be a tremendous risk of loss of articles, considering that researchers from all backgrounds (psychologists, sociologists and others) perform their studies towards samples of nurses. Also, nurses´ research is often directed towards their patients’ outcomes, disregarding their own performance and practice. This is why we have opted to include all studies regarding nurses´ interventions, regardless of whom conducted the study. I hope this is now clear and we ensure to include the rationale for our choice for the eligibility of the studies´ paragraph.

  1. Will inter-operator reliability be studied?

Congratulations on your work. Best regards.

Response – Thank you again for your question and compliment. If by the inter-operator reliability, you mean the level of agreement achieved between independent researchers who assess the studies selected for our research, then our answer is YES. We will ensure that whoever selects the studies (2 independent elements from the team) against the inclusion criteria is supported by a third element, allowing disagreements to be resolved through discussion. Therefore, we will ensure to acknowledge your suggestion and mention the level of agreement amongst the independent researchers who have assessed the articles.

We appreciate your time and recommendations, hoping that addressing all issues found, the manuscript may be finally eligible for publication. We will shortly re-submit the revised manuscript accordingly, as required.

Again, thank you.

Kind Regards

The authors

Round 2

Reviewer 2 Report

The authors (AA) have carefully addressed the reviewers' comments. Overall the changes made have improved the manuscript.